

# People in early stages of Parkinson's disease are able to intentionally reweight the use of visual information for postural control

Caio F. Cruz[1,2], Giovanna G. Genoves[2], Flávia Doná[3], Henrique B. Ferraz[3] and José A. Barela[4]

[1] School of Arts, Sciences and Humanities, Universidade de São Paulo, São Paulo, SP, Brazil
[2] Institute of Physical Activity and Sport Sciences, Universidade Cruzeiro do Sul, São Paulo, SP, Brazil
[3] Movement Disorder Section, Universidade Federal de São Paulo, São Paulo, SP, Brazil
[4] Institute of Biosciences, Universidade Estadual Paulista, Rio Claro, SP, Brazil

Corresponding author
Caio F. Cruz, caiofcruz@usp.br

## ABSTRACT

**Background:** Parkinson's disease (PD) leads to several changes in motor control, many of them related to informational or cognitive overload. The aim of this study was to investigate the influence of knowledge and intention on the postural control performance and on the coupling between visual information and body sway in people with and without PD standing upright.

**Methods:** Participants were 21 people with PD (62.1 ± 7.2 years), stages 1 and 2 (Hoehn & Yahr scale), under dopaminergic medication, and 21 people in the control group (62.3 ± 7.1 years). Participants stood upright inside a moving room, performing seven trials of 60 s. In the first trial, the room remained motionless. In the others, the room oscillated at 0.2 Hz in the anterior-posterior direction: in the first block of three trials, the participants were not informed about the visual manipulation; in the second block of three trials, participants were informed about the room movement and asked to resist the visual influence. An OPTOTRAK system recorded the moving room displacement and the participants' sway. The variables mean sway amplitude (MSA), coherence and gain were calculated.

**Results:** With no visual manipulation, no difference occurred between groups for MSA. Under visual manipulation conditions, people with PD presented higher MSA than control, and both groups reduced the sway magnitude in the resisting condition. Control group reduced sway magnitude by 6.1%, while PD group reduced by 11.5%. No difference was found between groups and between conditions for the coupling strength (coherence). For the coupling structure (gain), there was no group difference, but both groups showed reduced gain in the resisting condition. Control group reduced gain by 12.0%, while PD group reduced by 9.3%.

**Conclusions:** People with PD, under visual manipulation, were more influenced than controls, but they presented the same coupling structure between visual information and body sway as controls. People in early stages of PD are able to intentionally alter the influence of visual information.

## INTRODUCTION

Parkinson's disease (PD) is chronic and neurodegenerative, characterized by the loss of dopaminergic neurons in the substantia nigra, leading to decreased levels of dopamine in the striatum and changes in motor control (*Elbaz et al., 2016*). A debilitating and costly problem for many people with PD is the occurrence of falls, which often result in injury (*Bloem et al., 2001*; *Genever, Downes & Medcalf, 2005*) and contribute to poor quality of life (*Franchignoni et al., 2005*; *Hendred & Foster, 2016*). Usually, the occurrence of falls is associated to postural control deterioration.

The postural control system provides the ability to maintain postural orientation and stability depending on interactions between sensory systems and motor and cognitive processes (*Shumway-Cook & Woollacott, 2000*; *Woollacott & Shumway-Cook, 2002*). When young adults need to control their postural orientation simultaneously with a task that increases attentional demand and therefore requires cognitive effort, their postural control performance deteriorates (*Prado, Stoffregen & Duarte, 2007*; *Aguiar et al., 2014*; *Bucci, Ajrezo & Wiener-Vacher, 2015*). This suggests that postural control functioning requires attentional resources (*Andersson et al., 2002*). Additionally, postural control in older adults may be affected more than in younger adults by attention demands (*Shumway-Cook & Woollacott, 2000*; *Teasdale & Simoneau, 2001*; *Woollacott & Shumway-Cook, 2002*).

The cognitive influence in postural control was also demonstrated in more complex situations such as visual manipulation, using the moving room paradigm. In this case, young adults who were informed about the visual manipulation due to the room movement were able to reduce body sway induced by the visual information (*Freitas & Barela, 2004*). In contrast, information about the visual manipulation, due to discrete movement of a moving room, was not sufficient to produce body sway reduction in people with PD who still presented larger body sway magnitude than their peers without PD (*Bronstein et al., 1990*).

The different results from young adults and people with PD might be the need and involvement of attention when one tries to change the visual-motor coupling. The intention to resist a visual manipulation, avoiding or minimizing body sway, is more effective than only knowledge about the manipulation (*Aguiar et al., 2014*). However, such change in the visual-motor coupling required cognitive effort and the visual influence would increase when the intention to resist it was performed concomitantly with a cognitive task (count down from one hundred to zero in steps of three) (*Aguiar et al., 2014*). Limited results suggest that the same cognitive task did not alter body sway magnitude of people with PD standing upright (*De Souza Fortaleza et al., 2017*) and, therefore, there is the need to further investigate the involvement of attention and postural control in people with PD.

Considering that the coupling between visual information and body sway can be modified when the person is informed about the visual manipulation (*Freitas & Barela, 2004*; *Aguiar et al., 2014*) or is required to resist it (*Barela et al., 2009*; *Aguiar et al., 2014*) and considering that PD compromises attentional resources (*Dujardin et al., 2013*), the question that arises is: does attentional demand influence the use of visual information for postural control in people with PD? In addition, does PD influence the allocation of attention differently when the postural task involves intentional and voluntary control, in relation to the situation in which the postural task relies only on nondiscriminatory sensory information, therefore, without high level nervous system control?

Therefore, the aim of this study was to investigate the influence of knowledge and intention on the postural control performance and on the coupling between visual information and body sway of people in early stages of PD and people without PD standing upright. The hypothesis was that knowledge and intention would not alter the postural control performance and the visual-motor coupling in the visual manipulation conditions in people with PD but would alter in people without PD.

## MATERIALS AND METHODS

### Participants

Twenty one people with idiopathic PD (age: 62.1 ± 7.2 years, six females and 15 males), who were in the early stages of disease severity 1 and 2 on the Hoehn and Yahr scale (*Hoehn & Yahr, 1967*) and received dopamine replacement medication, and 21 healthy people paired by age and sex (control group, age: 62.3 ± 7.1 years, six females and 15 males) participated in this study. Participants with PD were recruited from Movement Disorders Unit of the São Paulo Hospital and the State Public Server Hospital and inclusion criteria were: (1) idiopathic PD diagnosed according to the Movement Disorder Society criteria (*Postuma et al., 2015*); (2) no neurological diseases, except for PD; (3) normal or corrected visual acuity; (4) lack of auditory losses; (5) no vestibular dysfunction; (6) no dyskinesia; (7) no freezing of gait diagnosis; (8) no cognitive impairment, evaluated by the Mini-Mental State Examination; (9) have not undergone ablative surgery of subcortical structures and (10) not participating in deep brain stimulation therapy. All these criteria were based upon previous evaluations performed in the aforementioned hospitals. Participants of the control group were recruited using personal contacts. Anthropometric and clinical information for both groups is presented in Table 1. All participants provided informed written consent, according to procedures approved by Research Ethics Committee of Cruzeiro do Sul University (protocol # 022-2016).

### Procedures

Participants were invited to visit the laboratory and asked to stand inside a moving room. This room was constituted by three walls (2 m length, 2 m width, 2 m height) and a ceiling and it has been employed in several previous studies (*Cruz et al., 2018*). This structure was mounted on wheels, that allowed the room to move in the anterior-posterior (AP) direction while the ground did not move. The walls had a pattern of light and dark vertical stripes. The room was moved by a servomotor mechanism previously described in

**Table 1 Anthropometric and clinical information of both Parkinson and control participants.**

|  | Parkinson group | Control group |
|---|---|---|
| Sample size | 21 | 21 |
| Sex (women/men) | 6/15 | 6/15 |
| Age (years) | 62.1 ± 7.2 (48–72) | 62.3 ± 7.1 (48–72) |
| Body mass (kg) | 76.7 ± 12.0 | 74.0 ± 11.9 |
| Height (m) | 1.62 ± 0.11 | 1.65 ± 0.07 |
| BMI (kg/m$^2$) | 29.2 ± 4.1 | 27.3 ± 4.1 |
| Disease duration (years) | 4.6 ± 3.4 (2–14) | – |
| H&Y | Stage 1: $n = 3$<br>Stage 2: $n = 18$ | – |
| MDS-UPDRS-III (maximum 132 points) | 23.7 ± 10.1 (7–43) | – |
| MMSE (maximum 30 points) | 27.5 ± 2.8 (20–30) | 29.0 ± 1.2 (25–30) |

Note:
BMI, body mass index; H&Y, Hoehn & Yahr scale; MDS-UPDRS-III, Movement Disorder Society Unified Parkinson's Disease Rating Scale-Part 3; MMSE, Mini Mental State Examination.

*Cruz et al. (2018)*. There were two fluorescent lights (20 W) on the ceiling of the room to always keep the same lighting.

An examiner asked participants to stand barefoot in an upright position as still as possible, placing their feet hip-width apart, and to keep their gaze on a target on the front wall of the room at 1 m. A total of seven trials, each lasting 60 s, were performed by each participant. In the first trial, the room was not moved and stayed stationary. The following trials were performed in two blocks, each with three trials, in which the room was translated (back and forward) with frequency of 0.2 Hz, amplitude displacement of 0.48 cm and peak velocity of 0.6 cm/s. In the first block, no information about the room movement was provided, ensuring that participants did not discriminate the visual manipulation. Before the second block, participants were informed about the room movement and that such visual manipulation induces body sway and then they were asked to resist its influence (not sway with the room).

In order to mask any possible auditory cues that emanated due to the movement of the room, a random sound (white noise) was provided by a massage vibrator behind the moving room throughout the experiment. Body and room position were captured by one infrared emitting diode (IRED), placed at the scapula level of the participant's back and another IRED on the front wall of the moving room, respectively. These IREDs were tracked by one OPTOTRAK™ camera block (Northern Digital Inc., Waterloo, Canada) at the sampling rate of 100 Hz.

## Data analysis

Mean sway amplitude (MSA), for both AP and medial-lateral (ML) directions, was obtained in the stationary room condition. In order to calculate MSA, a first-order polynomial and the average of the time series from each data point were subtracted, employing the Matlab detrend function. After, the standard deviation of the time series was calculated, indicating sway magnitude variability.

In the moving room conditions, sway variability was obtained also employing the MSA. In addition, the relationship between the room position and body sway was examined. Mean sway variability and the relationship between room and body sway were obtained only for the AP direction, because this was the direction that the room was oscillated. The relationship between room position and body sway was examined employing two variables: coherence and gain. Coherence was used to examine relationship strength between the room position and body sway. Coherence values close to one/zero indicated strong/weak relationship between the room and sway, respectively (*Dijkstra, Schöner & Gielen, 1994*; *Barela et al., 2000*). Gain was used to examine the spatial coupling structure between the room position and body sway. In this case gain indicates the relative magnitude influence of room movement on body sway. Gain was calculated as the absolute value of the Fourier transform of body sway divided by the Fourier transform of room position at the stimulus frequency. A gain value of 1 indicated that body displacement at the scapula level was the same as the moving room amplitude, and lower/higher values indicated that body sway amplitude was lower/higher than the moving room amplitude (*Polastri et al., 2012*).

Custom routines written in Matlab (The MathWorks, Inc., Natick, MA, USA) were employed for all calculations.

### Statistical analysis

First, normality and homogeneity of variance were confirmed. Then, the statistical analysis used parametric tests. For the trials with no visual manipulation, a multivariate analysis of variance (MANOVA) was performed using group as factor and the MSA, for both AP and ML directions, as dependent variables. For the trials with visual manipulation, three analyses of variance (ANOVA) (2 × 2) were performed, having group and task conditions, this last one treated as repeated measure, as factors and MSA, coherence and gain as dependent variables. Task conditions were: (1) no information about the room movement and (2) with information about the room movement and request to resist.

For all variables in the visual manipulation conditions, the average from the three trials at each task condition was used. When necessary, follow-up univariate analyses and Tukey's Honestly Significant Difference post hoc tests were performed, and the significant level was kept at 0.05.

## RESULTS

Participants with and without PD were able to maintain the upright stance in all experimental conditions.

### Postural control performance

When the room remained motionless, there was no difference in body sway magnitude between the groups in either direction (Fig. 1). MANOVA revealed no group effect, Wilks Lambda = 0.90, $F(2, 39) = 2.18$, $p = 0.13$, $\eta_p^2 = 0.10$.

When the room oscillated, visual manipulation induced higher body sway magnitude in both PD and control participants. Figure 2 depicts time-series of moving room
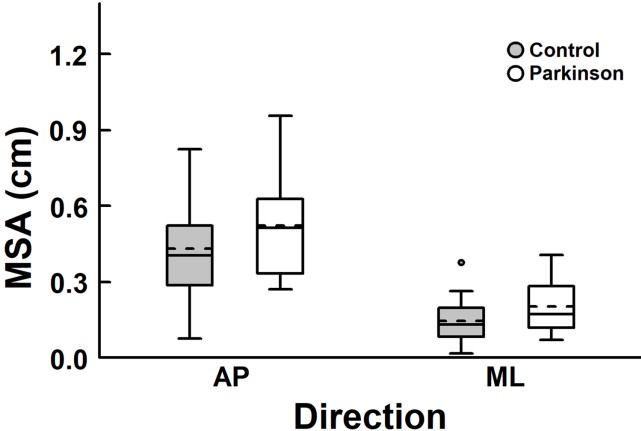

**Figure 1 Box plots of mean sway amplitude (MSA) with no visual manipulation, in the anterior-posterior (AP) and medial-lateral (ML) directions, for both PD and control groups.** The horizontal lines of the box plots indicate the lower extreme value, the 25th percentile (first quartile), the 50th percentile (median or second quartile), the 75th percentile (third quartile), and the upper extreme value. The dashed lines indicate the mean values. The circle indicates the outlier.

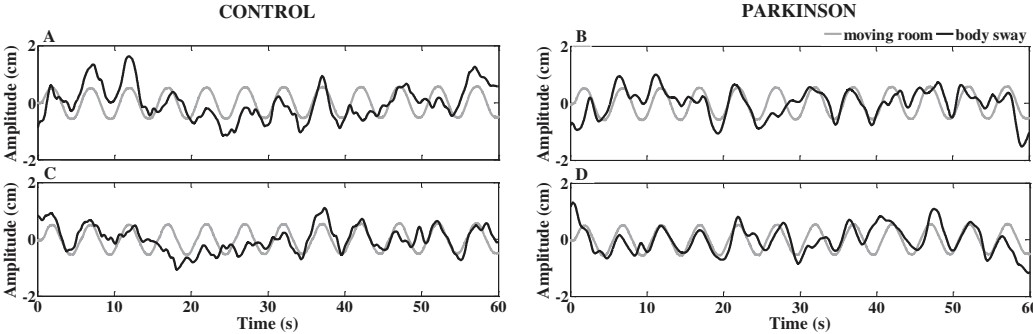

**Figure 2 Exemplar time-series of body sway and moving room displacements of a representative control and a representative Parkinson's disease participant, with no information about the room movement (A and B) and with information and request to resist (C and D).**

displacement and body sway from a representative control participant and a participant with PD.

Figure 3 depicts the MSA in the AP direction for both groups, in the trials with visual manipulation, and in the two experimental conditions. ANOVA revealed group, $F (1, 40) = 4.96$, $p = 0.032$, $\eta_p^2 = 0.11$, and condition effects, $F (1, 40) = 11.6$, $p = 0.002$, $\eta_p^2 = 0.22$, but no group and condition interaction, $F (1, 40) = 1.95$, $p = 0.17$, $\eta_p^2 = 0.047$. Participants with PD swayed with larger magnitude compared to their peers with no PD. In addition, when participants were asked to resist the visual manipulation, body sway magnitude was smaller compared to the condition where there was no information about the visual manipulation. Control group reduced sway magnitude by 6.1%, while PD group by 11.5%. In addition, 52.4 and 76.2% of the participants in the control and PD
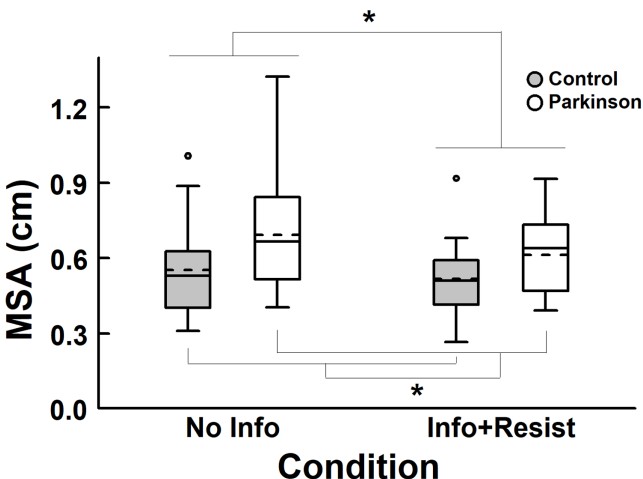

**Figure 3 Box plots of mean sway amplitude (MSA), in the anterior-posterior (AP) direction, for both PD and control groups, with no information about the room movement (No Info) and with information and request to resist (Info + Resist).** The horizontal lines of the box plots indicate the lower extreme value, the 25th percentile (first quartile), the 50th percentile (median or second quartile), the 75th percentile (third quartile), and the upper extreme value. The dashed lines indicate the mean values. The circles indicate the outliers. *Denotes significant difference $p < 0.05$.

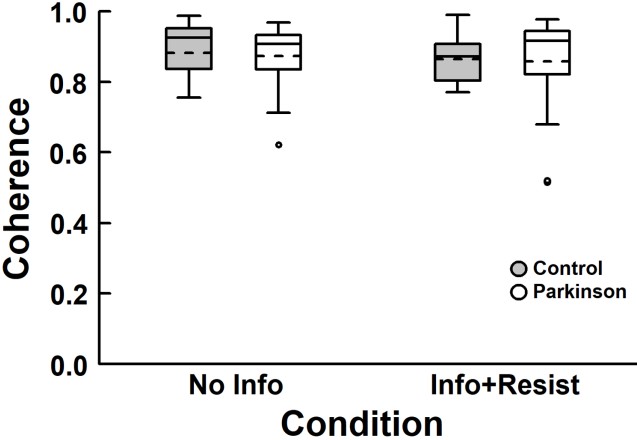

**Figure 4 Box plots of coherence for both PD and control groups, with no information about the room movement (No Info) and with information and request to resist (Info + Resist).** The horizontal lines of the box plots indicate the lower extreme value, the 25th percentile (first quartile), the 50th percentile (median or second quartile), the 75th percentile (third quartile), and the upper extreme value. The dashed lines indicate the mean values. The circles indicate the outliers.

group, respectively, reduced MSA when informed and asked to resist to the visual manipulation.

## Coupling between visual information and body sway

Figure 4 depicts coherence values between the room movement and body sway for both groups. ANOVA revealed no group, $F_{(1, 40)} = 0.11$, $p = 0.75$, $\eta_p^2 = 0.003$, and condition effects, $F_{(1, 40)} = 0.60$, $p = 0.44$, $\eta_p^2 = 0.015$, and no group and condition interaction,

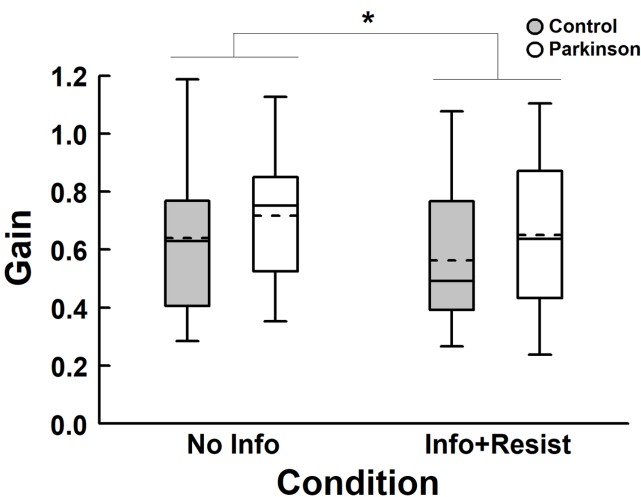

**Figure 5 Box plots of gain for both PD and control groups, with no information about the room movement (No Info) and with information and request to resist (Info+Resist).** The horizontal lines of the box plots indicate the lower extreme value, the 25th percentile (first quartile), the 50th percentile (median or second quartile), the 75th percentile (third quartile), and the upper extreme value. The dashed lines indicate the mean values. *Denotes significant difference $p < 0.05$.

$F (1, 40) < 0.001$, $p = 0.99$, $\eta_p^2 < 0.001$. Participants from both groups swayed coherently to the room in both conditions.

Figure 5 depicts gain between the room movement and body sway for both groups. ANOVA revealed no group effect, $F (1, 40) = 1.48$, $p = 0.23$, $\eta_p^2 = 0.036$, and no group and condition interaction, $F (1, 40) = 0.037$, $p = 0.85$, $\eta_p^2 = 0.001$, but revealed condition effect, $F (1, 40) = 7.23$, $p = 0.01$, $\eta_p^2 = 0.15$. In that case, the gain was lower when participants were informed about the room movement and asked to resist compared to the condition of no information. Control group reduced sway magnitude by 12.0%, while PD group reduced 9.3%. In addition, 66.7% of participants of both groups reduced gain values when informed and asked to resist to visual manipulation.

## DISCUSSION

The aim of this study was to investigate the influence of knowledge and intention on the coupling between visual information and body sway of people with and without PD standing upright. Our hypothesis was that knowledge and intention would not alter the postural control performance and the visual-motor coupling in people with PD but would alter in people without PD. In general, the results refute this hypothesis. First, knowledge about visual manipulation and the request to resist its influence altered postural control performance of people with PD, decreasing the magnitude of body sway (Fig. 3). Second, the visual-motor coupling was altered, decreasing the influence of the visual information on body sway, when participants were informed and asked to resist to the visual manipulation (Fig. 5). Moreover, gain reduction occurred for both groups, suggesting that people in stages 1 and 2 of PD changed the visual-motor coupling similarly to the control group.

When vision was not manipulated, there was no difference between groups (Fig. 1). In contrast, body sway magnitude was larger in the PD group compared to the control group, when visual manipulation occurred (Fig. 3). The different results from both visual conditions might be due to the PD early stages, as in the case of participants in this study, that might not have compromised postural control performance in less demanding conditions as in the case of when vision was not manipulated. Although still requiring further investigation, larger sway indicating poor postural control functioning, as observed in the visual manipulation conditions in this study, corroborates results from previous studies (*Horak, Nutt & Nashner, 1992*; *Horak, Dimitrova & Nutt, 2005*; *Doná et al., 2016*; *Cruz et al., 2018*), indicating that postural control in people with PD seems to be altered leading to a possible deterioration of its performance. Such different performance is not surprising because of the many changes that people with PD experience in both motor and sensory systems. Recently, we have demonstrated that larger magnitude sway occurs even when people with PD are exposed to visual manipulation in a continuous fashion (*Cruz et al., 2018*) indicating that postural control can couple and entrain to available cues, such as visual motion, but still the performance is worse compared to controls.

As previously mentioned, manipulation of the visual information, in conditions of small amplitude and low velocity, induces corresponding body sway without the person consciously discriminating the manipulation, as already previously observed (*Freitas & Barela, 2004*; *Barela et al., 2009*). Because participants were not perceptually aware of the visual manipulation, it might be suggested that the sensorimotor system operates in an automatic control model (*Stoffregen et al., 2006*) or, as recently suggested, in an intrinsic dynamic mode, which requires minimum, if any, involvement of higher centers of the Central Nervous System (CNS) to interpret changes in the environment and then plan and control the appropriate postural responses (*Genoves et al., 2016*). Automatic control is a useful mechanism because it allows the CNS to allocate attentional and cognitive resources to perform other tasks that are not related to postural control. Recent results showed that such a mechanism is intact and properly functioning in people with PD, allowing them to use the visual information to control their postural orientation while requiring low cognitive involvement, sparing attentional resources that could be directed to tasks that require voluntary effort (*Cruz et al., 2018*). Our results confirm this previously observation, showing that people with PD couple to continuous visual manipulation displaying corresponding body oscillation. Therefore, the intrinsic coupling between visual information and body sway in people with PD is intact.

The intrinsic sensorimotor coupling can be altered by previous knowledge, such as verbal information about visual manipulation, reducing body sway induced by visual stimulus (*Freitas & Barela, 2004*; *Barela et al., 2009*; *Aguiar et al., 2014*). Our results showed that people with PD can also alter the influence of visual information on body sway based upon knowledge and intention. When informed about the visual manipulation, gain values decreased compared to the no information condition. Besides observing that people with PD can down weight visual influence, our results showed no difference between people with and without PD. Considering that reduction in visual influence requires

attentional efforts (*Aguiar et al., 2014*; *Genoves et al., 2016*), our result suggests that people in early stages of PD can allocate attentional resources to alter sensory cue influences on postural control tasks. This possible important use of attention by people with PD is surprising and needs to be further investigated. It might be that because PD participants, in this study, were in early stages, they would have still the possibility of using attentional resources. However, such suggestion is speculative and needs to be taken cautiously. While the influence of visual information on induced body sway at the same stimulus frequency was reduced when participants were asked to resist, there was no change in the strength of the coupling between visual information and body sway, as indicated by the coherence values (Fig. 4). This result corroborates observations by *Barela et al. (2009)*. Based upon our results, we can suggest that people in early stages of PD can reduce but still are influenced by visual information, showing that the CNS cannot totally ignore available sensory cues even in situations where they provided unreliable information. The larger sway magnitude still observed in overall body sway in people with PD, as observed when visual manipulation occurred in this study, might be due to less accurate proprioceptive contribution to upright stance control. Recent studies examining sensory reweighting have shown proprioceptive deficit in people with PD (*Hwang et al., 2016*; *Feller, Peterka & Horak, 2019*) also observed in direct measures (G.G. Genoves, 2019, submitted; *Teasdale, Preston & Waddington, 2017*). Such lack of accurate proprioceptive cues could explain the increased reliance in visual manipulation not only observed in people with PD, as our results showed, but also in older adults, as previously shown (*Toledo & Barela, 2010*, *2014*). Thus, request to resist can down weight the influence of visual manipulation, but induced body sway still occurred in both people with PD and age-matched controls.

## CONCLUSIONS

The present study demonstrated that people in early stages of PD can use knowledge and intention to reduce the visual influence in maintaining upright stance similarly to their age-matched peers. Our observations indicate that people with PD can allocate attentional resources to change the automatic control properties by altering the use of sensory cues for postural control. The capability of sensory-motor reweighting is important in daily life situations because it allows people to change the influence of a particular, for instance unreliable, sensory source and to use other sensory cues during motor tasks, adapting to the conditions according to the task and environmental demands.

Future studies should explore the influence of attentional resources on the use of visual cues, for example, asking the participant to sway with the moving room, investigating adaptation and re-weighing mechanisms by manipulating sensory cues from other systems (e.g., somatosensory), including participants with PD in different stages and ages, and developing intervention protocols to improve sensory integration strategies.

## ACKNOWLEDGEMENTS

The authors thank Ana Maria Forti Barela, Douglas Vicente Russo-Junior, Tatiane Alessandra Miranda Andrade, and Vitor da Silva Amaral for assistance in data collection.

### Funding

This study was financed by the Coordenação de Aperfeiçoamento de Pessoal de Nível Superior—Brasil (CAPES)—Finance Code 001, and the São Paulo Research Foundation (FAPESP)—Grant #2016/06292-1. The funders had no role in study design, data collection and analysis, decision to publish, or preparation of the manuscript.

### Grant Disclosures

The following grant information was disclosed by the authors:
Coordenação de Aperfeiçoamento de Pessoal de Nível Superior—Brasil (CAPES)—Finance Code 001.
São Paulo Research Foundation (FAPESP): #2016/06292-1.

### Competing Interests

The authors declare that they have no competing interests.

### Author Contributions

- Caio F. Cruz conceived and designed the experiments, performed the experiments, analyzed the data, prepared figures and/or tables, authored or reviewed drafts of the paper, and approved the final draft.
- Giovanna G. Genoves conceived and designed the experiments, performed the experiments, analyzed the data, authored or reviewed drafts of the paper, and approved the final draft.
- Flávia Doná performed the experiments, analyzed the data, authored or reviewed drafts of the paper, selected the participants with Parkinson's disease, and approved the final draft.
- Henrique B. Ferraz conceived and designed the experiments, authored or reviewed drafts of the paper, selected the participants with Parkinson's disease, and approved the final draft.
- José A. Barela conceived and designed the experiments, analyzed the data, prepared figures and/or tables, authored or reviewed drafts of the paper, and approved the final draft.

### Human Ethics

The following information was supplied relating to ethical approvals (i.e., approving body and any reference numbers):

The Research Ethics Committee of Cruzeiro do Sul University approved the conduct and procedures of this study (protocol # 022-2016).

### Data Availability

The raw data is available as a Supplemental File.

## Supplemental Information

Supplemental information for this article can be found online at http://dx.doi.org/10.7717/peerj.8552#supplemental-information.

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
