# Peer review of "People in early stages of Parkinson’s disease are able to intentionally reweight the use of visual information for postural control"

_PeerJ, doi:10.7717/peerj.8552_

## Round 0.1 · original submission · Major Revisions

The reviewers commented positively on the manuscript and provided detailed and constructive feedback that the authors should use to further improve their manuscript. In particular, it has been recommended that data of individual participants are added to the figures showing group averages (https://doi.org/10.1371/journal.pbio.1002128) and free software is available to generate these plots showing (e.g. https://dx.doi.org/10.12688%2Fwellcomeopenres.15191.1). The authors should also report effect sizes.

·

Basic reporting

no comment

Experimental design

no comment

Validity of the findings

no comment

Additional comments

This is a clearly-written manuscript adding evidence about the effect of Parkinson's disease (PD) on body sway control. The Authors measured body oscillations while subjects were in an room moving in antero-posterior direction. Twenty-one patients with early PD performed similarly to 21 age-matched controls in all measures apart from oscillation amplitude, higher in PD patients either when trying to inhibit visual influence of the moving room or not. The author discuss that less accurate proprioception in PD patients could justify these results.
The study is sound, I only report some minor issues:
- Table 1: replace "High" with "Height" and replace commas with decimal dots

Reviewer 2 ·

Basic reporting

The English usage is slightly deficient as written. In the 'Minor Comments' of my comments to the authors I include suggestions/corrections that will correct these deficiencies.

By all other criteria for basis reporting, the article is acceptable.

Experimental design

The authors need to provide a reference for the frequency domain analysis described in the Methods section or they need to give a more detailed description (i.e., equations).

There is a possible mistake in the statement about the visual stimulus amplitude used in the experiments.

All other aspects of the experimental design are acceptable.

Validity of the findings

I offer suggestions in my Major Comments to the authors that ask for a more balanced interpretation of their results and some additional explanation.

The underlying processed data values (sway amplitude, gain, coherence) for each subject are provided. But I note that the spreadsheet with these data include an ID for each subject that appears to be subject initials. Typically it is not considered acceptable to include identifiers that can potentially reveal subject identity.

Additional comments

This study demonstrates that standing balance in subjects with early stage Parkinson’s disease (PD) is slightly more affected by visual disturbances caused by a moving room stimulus than age-matched controls, as judged by changes in the ‘mean sway amplitude’ (MSA, standard deviation of anterior-posterior body sway measured at upper torso level), but there was no statistical difference between PD and controls when response gain was used to compare subject groups. When subjects were asked to resist the influence of the visual stimulus, both PD and control subjects were able to reduce sway to a small extent as measured by both mean sway amplitude and gain measures. These results demonstrate that early stage PD subjects maintain the ability to consciously regulate sensory integration strategies that partially limit postural disturbances due to ‘false’ visual motion cues to an extent that is similar to the ability of age-matched controls. This might be considered a negative result, but it is a result worth making since it demonstrates an important functional characteristic of PD that is not affected or only minimally affected by the disease process.

Major Comments:

1. The authors appear to have interpreted their results correctly in terms of a standard statistical analysis, but what is missing is a more detailed description and representation of the magnitude of the changes observed between the Resist and the No Info conditions. Rather than just showing mean values and SD error bars in figures 3, 4, and 5 it would be more informative to show box plots that better represent the distributions of the data and show any outliers if they are present. Also, a useful addition to the statistical description of the various results related to figures 3 and 5 would be to state the percent reduction in MSA and gain for the two subject groups and to give the percentage of subjects in each group that actually show a reduction in MSA and gain between the No Info and the Resist conditions. Additionally, the abstract should be modified to give some information about the magnitude of changes observed.
2. The fact that the MSA results showed group differences but the Gain results did not is not an expected result, but there is no discussion about this. The simplest understanding of the results is that the stimulus-evoked sway would add to the spontaneous sway such that if spontaneous sway was the same and gain was the same in the two groups, then the stimulus-evoked MSA also should be the same. This reviewer suspects that this difference could be due to an accumulation of effects. Specifically, the AP MSA is slightly larger, on average, in PDs versus Controls (Fig. 1) and the Gain is slightly larger in PDs versus Controls Fig. 5). The combination of these two tendencies, even though neither effect reached statistical significance, may have combined to give a large enough group effect to reach statistical significance in the MSA results (Fig. 3). See also Minor Comment #17. This reviewer thinks the ‘big picture’ conclusion should emphasize the similarity between the PD and control groups rather than the small differences. Currently the Abstract does not covey this big picture view.
3. Line 136. The authors state that the peak-to-peak velocity of the stimulus was 0.6 cm/s. First, this amplitude should be stated in terms of peak displacement (cm) instead of velocity since all other measures in the paper are displacement measures. Second, are the authors certain the 0.6 cm/s is the peak-to-peak velocity and not the peak velocity? For a sinusoidal stimulus, the relation between peak displacement and peak velocity is Amp_disp = Amp_vel / (2 * pi * frequency). If the peak-to-peak velocity was 0.6 cm/s then Amp_vel was 0.3 cm/s and Amp_disp = 0.24 cm. The stimulus traces that are plotted in Figure 2 appear to show peak displacements of about 0.5 cm (not 0.24 cm) which makes me think that the peak stimulus velocity (and not peak-peak velocity) was 0.6 cm/s, which gives a peak amplitude of 0.48 cm. Or maybe the peak displacement amplitude was 0.6 cm as in the Barela et al., 2009 paper.
4. English usage needs some improvement – see numerous suggestions below.

Minor Comments:

1. Line 39. Suggest ‘. . . but both groups showed reduced gain in the . . .’
2. line 42. Suggest ‘. . . body sway as controls.’ In general, I suggest not using ‘peers’ as this is not as clear as saying ‘controls’ or ‘control group’.
3. Line 54,55. Suggest ending sentence with ‘deterioration’.
4. Sentence beginning on line 63. Suggest ‘Additionally, postural control in older subjects may be affected more than in younger subjects by attention demands.’
5. Line 69. Suggest substituting ‘In contrast’ for ‘Differently’.
6. Line 77. Should be ‘visual-motor’.
7. Line 78. Should be ‘. . . to resist it was . . .’
8. Line 79. Suggest ‘Limited results suggest that the same . . .’
9. Line 140. Should be ‘. . . asked to resist its influence . . .’
10. Line 141. How was the auditory masking applied? Through head phones?
11. Lines 150,151. Common understanding is that a ‘first order polynomial’ would be an equation (A + B*t) which is more clearly described as removing a linear trend.
12. Line 148. The procedures described in the data analysis section should include a reference that provides more details about the methods used. I note that a coherence value calculated for a frequency analysis gives information about the signal-to-noise ratio of the stimulus-response data at the stimulus frequency and not specifically the ‘dependency’ between the stimulus and response. One could have a constant and unchanging dependency of the response on the stimulus, but if there is additional uncorrelated variability in the response signal (e.g. due to internal noise in the system) then the coherence would be reduced. See Pintelon & Schouten textbook on frequency domain approach to system identification (Wiley-IEEE press).
13. Sentence beginning on line 187. This sentence is not necessary.
14. Discussion section. In general, when results are described and summarized, the authors should include references to the specific figures were those results are shown.
15. Line 255. Should be ‘. . . refute this hypothesis.’
16. Line 256. Should be ‘. . .to resist its influence . . .’
17. Line 263. Suggest ‘In contrast’ instead of ‘Differently’. But more importantly, regarding the point being made here that there was a group difference with visual stimulation but no difference without visual stimulation, this statement is relying too much on binary classification based on statistical significance while ignoring trends in the results. Also, I think your statistical analysis that determined that there was a group difference with visual stimulation relied on combined results from the No Info and Resist conditions (and both of these trials three repeats), so this analysis had a repeated-measures advantage in detecting a difference that was not available in the non-stimulated condition. Please provide a description that is fairer to your data.
18. Line 264. Not clear what ‘This observation’ is referring to. The previous sentence was saying that there were no differences between groups while this sentence is saying that PD subject posture is poorer.
19. Line 269. Should be ‘. . . with PD are exposed . . . ‘
20. Line 217. Suggest ‘. . . such as visual motion, . . .’ and also use ‘controls’ rather than ‘peers’.
21. Line 273. Not clear what you mean by ‘without the person discriminate such manipulation’. Do you mean that the subject is not consciously or perceptually aware that the visual scene was moving? Was your stimulus at this low, below conscious level? This argument about what does and does not require ‘higher centers’ is a bit tricky since an automatic control mode (I think ‘automatic control mode’ would be a better descriptor than ‘intrinsic dynamic mode’ in line 275, and also who’s to say that automatic control does not require ‘higher centers’ – this reviewer thinks it does) can be operative whether the subject is aware or not of the stimulus.
22. Line 278. Better to say ‘That is a useful mechanism . . .’
23. Line 284. Should be ‘. . . displaying corresponding body oscillation.’
24. Lines 285,286. Suggest ‘. . . body sway in people with PD is intact.’. . .
25. Line 309. Should be ‘. . . explain the increased reliance . . .’
26. Line 310. Should be ‘. . . as previously shown . . .’
27. Lines 312,313. Suggest ‘. . . but induced body sway still occurred in both people with PD and age-matched controls.’
28. Sentence beginning line 313. This sentence should be moved to the next section (future studies) if you want to keep it.
29. Line 319. Should be ‘. . . to reduce the visual . . .’
30. Line 323. Eliminate the word ‘accordingly’
31. Sentence beginning on line 325. Eliminate this last sentence since it is repeating things already said.
32. Lines 330-333. The last part of this sentence is not clear. Suggest ‘. . . developing intervention protocols to improve sensory integration strategies.’
33. Figure legend 2 should refer to '(A and B)' and '(C and D)'.
34. The raw data provided in the spreadsheet should not use subject initials as identifiers since this can potentially be used to identify the subjects.

Reviewer 3 ·

Basic reporting

The paper investigated the influence of knowledge and intention on the postural control performance and on the coupling between visual information and body sway in people with and without PD standing upright. This is an original paper with interest findings. However, some points need to be improved.

line 200 - Induced what? Higher?

Table 1: Please change comma for point.

Please improve the resolution of figure 2.

Experimental design

Inclusion and exclusion criteria: Please clarify how was verified each criterion.

Validity of the findings

discussion: “Differently, when vision was not manipulated, there was no difference between groups.”
This was no expected and should be discussed.

discussion: “our result suggests that people in the early stages of PD can allocate attentional resources to alter sensory cue influences on postural control tasks.”
Why are they able to allocate attentional resources similar to the control group? This needs to be better explained.

Additional comments

The paper investigated the influence of knowledge and intention on the postural control performance and on the coupling between visual information and body sway in people with and without PD standing upright. This is an original paper with interest findings. However, some points need to be improved.

1) Abstract and conclusion: “ … suggesting that they have and/or allocate attentional resources to promote such changes”.
It is not possible to conclude about the attentional resource with the findings of the study. I suggest to remove this suggestion or rewrite. Please look to in the study conclusion too.

2) Introduction: “its diagnosis is initially made by three motor signs: bradykinesia, rigidity and/or rest tremor (Postuma et al., 2015).”
This affirmation is not correct. The diagnostic of the disease occurs the different ways. If the authors are considering the differential diagnostic, there are other aspects that are considered beyond these three motor signs/symptoms. Please rewrite this sentence.

3) Introduction: paragraphs 3 and 4 do not interact. Please improve the writing.

4) Introduction: “Although sparse, results indicate that the same cognitive task did not alter body sway magnitude of people with PD standing upright (Fortaleza et al., 2017).”
Please double-check this information. I’m not sure that it is possible to affirm this.

5) hypothesis: It is not clear the justification for this hypothesis. If the people with PD presents the attentional resources compromised, is not expected that knowledge and intention, which increase the attentional demand, impair the postural control performance?

6) Inclusion and exclusion criteria: Please clarify how was verified each criterion.

7) line 200 - Induced what? Higher?

8) discussion: “Differently, when vision was not manipulated, there was no difference between groups.”
This was no expected and should be discussed.

9) discussion: “our result suggests that people in the early stages of PD can allocate attentional resources to alter sensory cue influences on postural control tasks.”
Why are they able to allocate attentional resources similar to the control group? This needs to be better explained.

10) Table 1: Please change comma for point.

11) Please improve the resolution of figure 2.

---

## Round 0.2 · Minor Revisions

The reviewers are satisfied with the revised manuscript but have suggested a number of textural improvements that should be addressed before the manuscript can be accepted for publication.

·

Basic reporting

no comment

Experimental design

no comment

Validity of the findings

no comment

Additional comments

I raise no further comments

Reviewer 2 ·

Basic reporting

This reviewer is satisfied with the authors’ responses to the reviewers and with the revisions made to the manuscript. I only offer a number of suggestions to improve the English usage. These are listed in the general comments to the authors. None of these changes are substantial enough to require additional review.

Experimental design

All criteria are met.

Validity of the findings

All criteria are met.

Additional comments

Minor comments:

1. Line 56. Suggest: ‘The postural control system provides the ability to . . .’
2. Line 75. Suggest: ‘resist a visual manipulation . . .’
3. Line 82. Add period at end of the sentence.
4. Line 122. Suggest: ‘This structure was . . .’
5. Line 132. Suggest: ‘. . . the room was translated (back and forward) . . .’
6. Lines 152, 153. Suggest: ‘In addition, the relationship between the room position and body sway was examined. Mean sway variability . . .’
7. Line 156. Suggest: ‘Coherence was used . . .’
8. Line 159. Suggest: ‘Gain was used . . .’
9. Lines 160-163. Suggest: ‘In this case gain indicates the relative magnitude influence of room movement on body sway. Gain was calculated as the absolute value of the Fourier transform of body sway divided by the Fourier transform of room position at the stimulus frequency. A gain value of 1 indicated that body displacement at the scapula level was the same as the moving room amplitude, and . . .’
10. Lines 166,167. Suggest: ‘Custom routines written in Matlab (The MathWorks, Inc.) were employed for all calculations.'
11. Line 202. Suggest: ‘. . .to resist the visual manipulation, . . .’
12. Line 203. Suggest: ‘. . .the condition where there was . . .’
13. Line 210. Suggest: ‘. . .and condition effects, . . .’
14. Line 219. Suggest: ‘. . . compared to the condition . . .’
15. Lines 241, 242. Suggest: ‘. . . in the case when vision was not manipulated.’
16. Line 247. Suggest: ‘. . . because of the many changes . . .’
17. Line 254. Suggest: ’. . . discriminating the manipulation . . .’
18. Line 255. Suggest: ‘. . . not perceptually aware of the visual . . .’
19. Line 260. Suggest: ‘Automatic control is a useful mechanism . . ‘
20. Line 262. Suggest: ‘. . . that such a mechanism . . .’
21. Lines 263, 264. Suggest: ‘. . . postural orientation while requiring low . . .’
22. Line 272. Suggest: ‘. . . on body sway when given . . .’
23. Line 280: Suggest: ‘. . . would have still the possibility . . .’
24. Line 282: Suggest: ‘While the influence of visual information on induced body sway was reduced . . .’
25. Line 285. Suggest: ‘. . . corroborates observations by . . .’
26. Lines 287, 288. Suggest: ‘. . . cannot totally ignore available sensory cues even in situations where they . . .’
27. Lines 304, 305. Suggest: ‘Our observations indicate that people with PD can allocate attentional resources to change the automatic control properties by altering the use of sensory cues for postural control.’
28. Line 310. Suggest: ‘. . . explore the influence of attentional resources on the use of visual cues, . . .’
29. Figure 1 legend. This figure legend should define the parameters of the boxplots. There is no standard for the representation of box plots. Presumably one of the lines within the box is a median value and the other a mean value. Maybe the box represents the 25 and 75 percentile points, and maybe the whiskers are 5 and 95 percentiles.

---

## Round 0.3 · accepted · Accept

The authors have adequately addressed the remaining comments.